# Prevalence of Enterococci and Vancomycin Resistance in the Throat of Non-Hospitalized Individuals Randomly Selected in Central Italy

**DOI:** 10.3390/antibiotics12071161

**Published:** 2023-07-07

**Authors:** Annalisa Palmieri, Marcella Martinelli, Agnese Pellati, Francesco Carinci, Dorina Lauritano, Claudio Arcuri, Luigi Baggi, Roberto Gatto, Luca Scapoli

**Affiliations:** 1Department of Medical and Surgical Sciences, University of Bologna, 40138 Bologna, Italy; annalisa.palmieri@unibo.it (A.P.); luca.scapoli2@unibo.it (L.S.); 2Department of Translational Medicine and for Romagna, University of Ferrara, 44121 Ferrara, Italy; agnese.pellati@unife.it (A.P.); francesco.carinci@unife.it (F.C.); dorina.lauritano@unife.it (D.L.); 3Department of Clinical Sciences and Translational Medicine, University of Rome “Tor Vergata”, 00113 Rome, Italy; arcuri@med.uniroma2.it (C.A.); luigi.baggi@uniroma2.it (L.B.); 4Department of Life, Health and Environmental Sciences, School of Dentistry, University of L’Aquila, 67100 L’Aquila, Italy; roberto.gatto@univaq.it

**Keywords:** *Enterococcus faecalis*, *Enterococcus faecium*, vancomycin, prevalence, resistance, antibiotics, oral cavity, screening

## Abstract

Enterococci are commonly found in the environment and humans as a part of the normal microbiota. Among these, *Enterococcus faecalis* and *Enterococcus faecium* can convert into opportunistic pathogens, making them a major cause of nosocomial infections. The rapid diffusion of vancomycin-resistant strains and their impact on nosocomial settings is of considerable concern. Approximately one-third of the *E. faecium* infections in Italy are caused by vancomycin-resistant strains. This study explored the hypothesis that the oral cavity could represent a silent reservoir of virulent enterococci. A sample of 862 oral flora specimens collected from healthy human volunteers in Central Italy was investigated by real-time PCR to detect *E. faecalis* and *E. faecium*, as well as the genetic elements that most frequently determine vancomycin resistance. The prevalence of *E. faecalis* was 19%, a value that was not associated with alcohol consumption, tobacco smoking, or age of the subjects. Less frequently detected, with an overall prevalence of 0.7%, *E. faecium* was more common among people older than 49 years of age. The genes conferring vancomycin resistance were detected in only one sample. The results indicate that the oral cavity can be considered a reservoir of clinically relevant enterococci; however, our data suggest that healthy individuals rarely carry vancomycin-resistant strains.

## 1. Introduction

Enterococci are ubiquitous Gram-positive bacteria in nature and commensals in humans, constituting approximately 1% of the fecal microbiota [1]. The species *Enterococcus faecalis* and *Enterococcus faecium* are the most common in humans; they are commensals in the oral cavity, intestine, and genitals, but can become opportunistic pathogens causing bacteriemia, endocarditis and meningitis, as well as wound, soft tissue, prosthetic and urinary infections. *E. faecalis* is also involved in periodontitis, periimplantitis and endodontic infections [1,2,3]. Enterococci are naturally resistant to many classes of antibiotics, and resistance is often acquired by the selective pressure caused by the inappropriate use of antibiotics. As the use of a given antibiotic significantly reduces its efficacy, infections previously eliminated with ease are joined by others that require more attention to eradicate, further worsening the health of masses of individuals. This is happening in India, where antibiotic resistance causes more deaths than cancer and traffic accidents combined [4]. More and more attention is being paid to the drastic reduction in the effectiveness of one of the most important discoveries of the 20th century [5], which is causing a multifaceted public health issue recognized by the World Health Organization (WHO). Specifically, vancomycin-resistant enterococci (VRE), a frequent cause of epidemics, have been declared as “high priority” and “serious treat” pathogens by the WHO. About 20 thousand deaths were caused by VRE worldwide in 2019 [6].

Vancomycin was isolated from *Streptomyces orientalis* in 1957. It is a glycopeptide that inhibits the wall synthesis of Gram-positive bacteria [7]. The low toxicity of the drug and its effectiveness in the treatment of methicillin-resistant *Staphylococcus aureus* (MRSA) epidemics have favored both the adoption and widespread use of vancomycin.

The molecular mechanism of acquiring vancomycin resistance is linked to the presence of operons that can be horizontally transferred by mobile genetic elements. Although there are a variety of operons that confer vancomycin resistance in enterococci, only the *vanA* and *vanB* types are widespread, and thus they receive greater public health attention [8]. The operon *vanA* is found on the transposon Tn1546 and takes its name from one of its seven genes, the gene of the ligase *vanA* [9]. The *vanB* operon, characterized by *vanB* ligase, is often integrated into the host genome [10]. The prevalence of *vanA* and *vanB* types varies worldwide; *vanA* is more common in North America and Europe, while *vanB* is common in Oceania and is increasing in Europe [11,12,13]. Mobile elements can often simultaneously transfer multiple antibiotic resistances. This complicates the clinical treatment of infections and management of surveillance and containment programs.

Clinically relevant bacterial species that frequently exhibit vancomycin resistance include *E. faecium*, *E. faecalis*, and *S. aureus*. According to the Surveillance Atlas of the Infectious Diseases European Centre for Disease Prevention and Control website (http://atlas.ecdc.europa.eu/public/index.aspx, accessed on 25 February 2023), the prevalence of VRE is increasing rapidly. In particular, the percentage of vancomycin-resistant *E. faecium* infections has more than doubled in the last five years, reaching 28.2% in 2021. Vancomycin resistance in *E. faecalis* and *S. aureus* was rarer, with values of approximately 1% in Italy.

Acquired vancomycin resistance is most prevalent among enterococci and is still rare in other pathogenic bacteria, such as *S. aureus* and *Clostridium difficile* [7,14]. Although the number of cases of vancomycin-resistant *S. aureus* (VRSA) infection is limited, this poses a potential threat to public health, because vancomycin is a key bactericidal drug used in the treatment of invasive MRSA infection. It was observed that most of the VRSA strains acquired vancomycin resistance by the transposon Tn1546 from *E. faecalis* [7]. The successful transfer of the van element from *E. faecalis* to an MRSA strain in a mixed infection was confirmed in vitro and in mice [15]. 

Nosocomial infections represent only a part of the field focusing on the containment of antibiotic resistance. The zootechnical field is equally relevant, where the wide use of antibiotics generates strong positive selective pressure and could allow for the spread of resistant bacteria to the general population through foodstuff [16,17,18]. Agriculture also contributes via the selective pressure of resistant bacterial species [19]. In fact, through fertilizers, groundwater and surface runoff, vegetables and fruits used for consumption receive up to 90 percent of the antibiotics administered to farm animals [20].

The general population may represent a significant reservoir for bacterial species and virulent strains that need to be monitored. An increasing number of reports claim that enterococci routinely inhabit the oral cavity [21,22]. Among these, *E faecalis* plays a pathogenic role because it seems to also be involved in periodontitis, periimplantitis, and endodontic infections [2,3], while the percentage of vancomycin-resistant *E. faecium* infections was reported to have more than doubled in the past five years. Interestingly, it was suggested that oral plaque could be an environment that facilitates horizontal gene transfer and the spread of antibiotic resistance genes among biofilm inhabitants [21]. These elements provide a rationale for investigating the prevalence of *E. faecalis* and *E. faecium* in a large sample of healthy Italian subjects and monitoring the presence of vancomycin-resistance genes to understand whether the oral cavity could represent a silent reservoir of virulent enterococci.

## 2. Results

A sample of 862 participants was investigated to detect sequences of *E. faecium*, *E. faecalis* and operons that confer vancomycin resistance of the *vanA* and *vanB* types. Biological specimens obtained from throat swabs were selected and validated in a previous study [23]. The sample study was randomly selected and not representative of the population; indeed, it was enriched in females (61%) and young adults. As expected, *E. faecalis* was found more frequently than *E. faecium*. The overall prevalence of these two species was 0.185 (95% C.I. 0.161–0.213) and 0.007 (95% C.I. 0.003–0.014), respectively. The prevalence, stratified by age group, is shown in Figure 1. 

The age class 11–20 showed a lower prevalence of *E. faecalis*, but the difference was not significant with respect to the rest of the sample (*p* value = 0.06). Five out of six *E. faecium*-positive samples came from people older than 49 years; the prevalence of *E. faecium* in more senior volunteers (0.0192) was significantly higher than that in younger people (0.0017; *p* value = 0.005).

The occurrence of *E. faecalis* and *E. faecium* was not associated with sex, smoking habits, or alcoholic beverage consumption (Table 1 and Table 2).

The sequence of the *vanA* gene was not detected, whereas only one sample was positive in the *vanB* assay. Interestingly, this sample tested negative for both *E. faecalis* and *E. faecium*.

## 3. Discussion

Enterococci are both common commensals and major opportunistic human pathogens. *E. faecalis* and *E. faecium*, the most common species in humans, frequently cause nosocomial infections. Since the fraction of VRE infections in Italy is rapidly increasing, we evaluated whether a healthy population could represent a silent reservoir of these virulent species. The oral cavity can be one of the principal anatomical sites colonized by enterococci; the relative abundance of the *Enterococcus* genus in the oral microbiome is 1.3% according to data by the Human Oral Microbiome Database [24]. A sample of 862 non-hospitalized volunteers was tested to detect *E. faecalis-* and *E. faecium*-specific DNA sequences, as well as sequences of *vanA* and *vanB* genes conferring vancomycin resistance. The overall prevalence of *E. faecalis* in oral specimens was 18.5%. This value was similar to that observed in Brazil, where 17% of the samples were positive for enterococci, mostly *E. faecalis* [25], or in the USA, where 20% of patients with healthy periodontium tested positive for *E. faecalis* [26]; however, it was much lower than that found in periodontal patients (70%) or in insulin-treated diabetics [27]. The prevalence of *E. faecalis* in our sample was not influenced by alcohol consumption or smoking. Age was also not influential, in contrast to the results observed in the Brazilian cohort, where an increasing degree of carriage in adults and the elderly was observed [25]. 

The prevalence of *E. faecium* in the present study was 0.7%. Little data regarding the oral prevalence of this bacteria in healthy subjects have been published. In the above-mentioned Brazilian study, *E. faecium* was 50 times rarer than *E. faecalis*, being detected only 2 times in 240 subjects. Enterococci were never detected among 30 healthy controls from India, but were common in the subgingival biofilm of patients with chronic periodontitis, where *E. faecium* was found in 10% of patients and *E. faecalis* in 85% of patients [28].

One limitation of this study was that the sample was not representative of the population because it was randomly selected. Therefore, the calculated overall enterococcal prevalence may differ from the actual population prevalence. However, as age seems to be one of the major factors influencing the prevalence, the data reported in Figure 1 may better resemble the real population distribution.

The screening for vancomycin resistance identified only one sample that was positive for the *vanB* type and none that were positive for the *vanA* type. The *vanA* and *vanB* genes characterize the mobile genetic elements responsible for acquired vancomycin resistance in clinically relevant enterococci. In several European countries, *vanA* enterococci have been isolated from the community and from sewage, feces from farm animals, and raw meat for human consumption, acquired at retail stores [16,29,30,31]. Intestinal colonization with VRE in the healthy population has been reported in Belgium, Morocco, and Taiwan at a rate of >20% [32,33,34]. On the other hand, no VRE was found in healthy Iranian children [35]. Unlike these studies, which examined stool samples, our study investigated samples from the oral cavity. We found a low prevalence of *E. faecium* in this anatomical region. Considering that this species is the most relevant among the VRE species, this fact may partially explain our results. Overall, we can conclude that the oral cavity of healthy subjects with good oral hygiene is not a significant reservoir for VRE. Previously published data agreed with this conclusion. Only three oral specimens from 879 dental students were found to be positive for VRE [36]. Our results were corroborated by a study in which *vanA* was not found in a small sample of 20 healthy individuals and 20 periodontitis patients [37].

In the present study, the detection of enterococci and vancomycin resistance was conducted using PCR. Compared to traditional cell culture methods, the molecular detection of DNA is more suitable for large-scale screening projects because it is faster and less expensive. Following comparison, PCR appeared to be more sensitive, possibly because it can detect DNA also from dead cells [25,26]. Traditional microbiological methods can provide a precise phenotyping of the cultured species. This could be an advantage, particularly for antibiotic resistance. In this study, a single sample carrying the vancomycin-resistance *vanB* gene was detected; however, this sample was negative for *E. faecium* and *E. faecalis*. Other species can acquire vancomycin-resistance operons, for instance, *S. aureus* and *C. difficile*; however, without culture phenotyping, it is impossible to discriminate between different hypotheses.

Overall, this investigation indicates that the oral cavity can be considered a reservoir of clinically relevant enterococci; indeed, *E. faecalis* appears to be relatively common. However, the results suggest that healthy individuals rarely carry *E. faecium* and vancomycin-resistant strains.

## 4. Materials and Methods

### 4.1. Sample Collection

The research sample included 862 oral flora specimens collected from healthy human volunteers. Subjects were randomly selected from the dental clinics of the University of Tor Vergata in Rome (Italy) and University of L’Aquila in L’Aquila (Italy) between December 2017 and March 2019. All enrolled subjects signed an informed consent form before sample collection. Parents signed an additional consent form for the participating children. The L’Aquila Ethics Committee approved this study (approval number 26/2017).

Patients with systemic disease, facial trauma, those who had undergone radio and/or chemotherapy, or those with poor oral hygiene were also excluded.

A sample of oral flora swabbing the surface of the tonsils and oropharynx was collected from each subject. The swabs were immediately placed in a test tube containing silica gel capsules to dry the specimens, which were then briefly stored at 4 °C until DNA purification. In brief, cells were lysed in two steps using lysozyme and proteinase K; then, DNA was automatically purified on silica filters using the QIAcube HT instrument and the QIAamp 96 DNA QIAcube HT Kit (Qiagen, Hilden, Germany).

### 4.2. Molecular Analysis

*E. faecalis*, *E. faecium*, *vanA*, and *vanB* genes were detected using real-time quantitative polymerase chain reaction (real-time qPCR). The absolute quantification assay for each specific target was performed using hydrolysis probes in the ABI PRISM 7500 thermal cycler (Applied Biosystems, Foster City, CA, USA). Highly specific primer–probe sets were designed with the Primer-BLAST online tool [38]. The primer quality was further checked with MFEprimer v3.0 software, which also helped to set the multiplex PCR assays [23]. Oligonucleotides were synthesized by Biomers.net (Ulm, Germany), and their sequences are reported in Table 3.

Each 20 µL reaction contained 10 µL of 2× qPCRBIO Probe Mix Lo-ROX, 100 ng of template DNA purified from samples, 200 nM of each primer, and 100 nM fluorescent probe. The thermal protocol started with 10′ at 95 °C to activate the polymerase, followed by 40 cycles of two-step amplification: 15″ at 95 °C and 60″ at 57 °C. Cloned synthetic DNA target sequences (Eurofin MWG Operon) were used as standards for the quantitative analysis. Serial dilutions of plasmids were used to set PCR reactions with a scalar number of target copies, from 10 to 10,000,000, to check the amplification efficiency and to quantify the sample by comparison with the standard curves, that is, threshold cycle values against the log of the copy number.

### 4.3. Statistical Analysis

Data analysis and descriptive statistics were performed using SPSS software v.25 (IBM, New York, NY, USA). Associations between variables were analyzed using two-way contingency tables and Pearson’s chi square test; two-sided *p* values < 0.05 were considered significant. The level of association between variables were evaluated using the odds ratio (OR). Prevalence confidence limits were calculated with the binomial proportions tool of the OpenEpi web site (www.openepi.com, accessed on 2 April 2023).

## Figures and Tables

**Figure 1 antibiotics-12-01161-f001:**
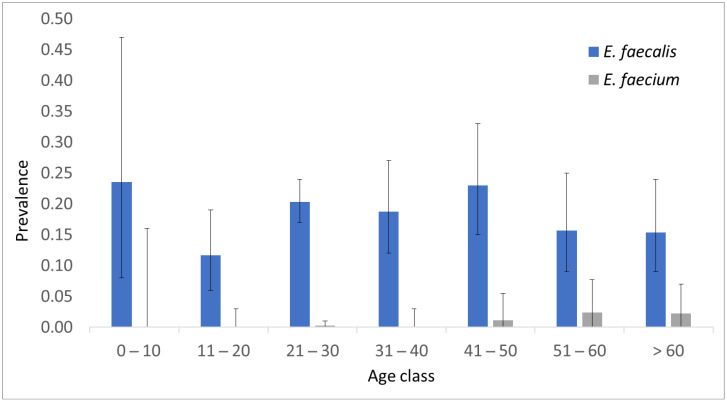
Prevalence distribution of *E. faecalis* (blue bars) and *E. faecium* (gray bars) in healthy population by age class. Error bars represent the 95% confidence interval for simple proportion.

**Table 1 antibiotics-12-01161-t001:** Occurrence of *E. faecalis* stratified by sex, smoking habits, and alcohol consumption.

		*E. faecalis*		*p* Value	OR (95% C.I.) ^1^
		(−)	(+)		
**sex**	male	281	53	0.10	1.35 (0.94–1.95)
	female	420	107		
**smoking**	(−)	466	102	0.51	1.13 (0.78–1.61)
	(+)	235	58		
**alchool**	(−)	433	104	0.43	0.87 (0.60–1.24)
	(+)	269	56		

^1^ OR = odds ratios; C.I. = confidence interval.

**Table 2 antibiotics-12-01161-t002:** Occurrence of *E. faecium* stratified by sex, smoking habits, and alcohol consumption.

		*E. faecium*		*p* Value	OR (95% C.I.) ^1^
		(−)	(+)		
**sex**	male	332	2	0.78	1.27 (0.22–9.96)
	female	523	4		
**smoking**	(−)	563	5	0.37	0.39 (0.02–2.80)
	(+)	292	1		
**alchool**	(−)	533	4	0.82	0.82 (0.11–4.68)
	(+)	323	2		

^1^ OR = odds ratios; C.I. = confidence interval.

**Table 3 antibiotics-12-01161-t003:** Oligonucleotide sequences for real-time PCR.

Target	Primer (5′–3′)	Probe (5′–3′)
*E. faecalis*	F-GGCATAAGAGTGAAAGGCGC	JOE-TTTCGTGTCGCTGATGGATGGACCCG-BHQ1
	R-CATCGTGGCCTTGGTGAG	
*E. faecium*	F-ACATGCAAGTCGAACGCTTC	6_FAM-TGCTCCACCGGAAAAAGAGGAGTGGCGA-BMN_Q535
	R-TACCCACGTGTTACTCACCC	
*vanA*	F-TTCATCAGGAAGTCGAGCCG	6_FAM-CCCGCAGACCTTTCAGCAGAGGAGCGA-BMN_Q535
	R-TGCCGTTTCCTGTATCCGTC	
*vanB*	F-ATTGAGCAAGCGATTTCGGG	CY5-TGTGAGGTCGGCTGCGCGGTCATGGGA-BMN_Q620
	R-TCCACTTCGCCGACAATCAA	

## Data Availability

The data used to support the findings of this study are available from the corresponding author upon request.

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
