# Peer review of "Prevalence of Enterococci and Vancomycin Resistance in the Throat of Non-Hospitalized Individuals Randomly Selected in Central Italy"

_antibiotics, 2023, doi:10.3390/antibiotics12071161_

Round 1
Reviewer 1 Report
This paper reports the results of “Prevalence of Enterococci and Vancomycin Resistance in the Throat of Non-Hospitalized Individuals Randomly Selected in Central”. In my opinion, the manuscript is in a position to be accepted for publication after major revision. Here are some comments on the manuscript:
- I am sorry to say that the problematic is not clear.
- The subject of the paper is not new and the results do not add novelty in the field
- The introduction should be less long and more concise
- I think that results do not respond to the problematic
Reviewer 2 Report
The manuscript analyzes the presence of the two main species of the Genus Enterococcus in oropharyngeal specimens from healthy patients including the analysis of basic clinical data.
It provides interesting information on the composition of the microbiota of the healthy population, analyzing a high number of microorganisms.
In order to improve the quality of the article, the discussion should be revised to include data from the latest literature on the analysis of the microbiome in healthy patients.
Reviewer 3 Report
In this short communication paper “Prevalence of Enterococci and Vancomycin Resistance in the Throat of Non-Hospitalized Individuals Randomly Selected in Central Italy”, Palmieri et al., investigates the hypothesis that the oral cavity could be a reservoir of virulent enterococci, including vancomycin-resistant strains. The study collected oral flora specimens from 862 healthy human volunteers in Central Italy and used real-time PCR to detect E. faecalis and E. faecium, as well as the genetic elements that most frequently determine vancomycin resistance. The study concludes that the oral cavity can be considered a reservoir of clinically relevant enterococci, but healthy individuals rarely carry vancomycin-resistant strains.
In general, the article is well-written and provides informative study on the prevalence of Enterococci and vancomycin resistance in the oral cavity of healthy individuals.
Here are some of the comments and suggestions for each section of the study:
1. Sample Selection:
As the study samples were randomly selected from dental clinics, it cannot be considered as representative of the general population. The samples from females and young adults are more enriched, making the results more skewed. The authors should consider selecting a more representative samples in future studies.
2. Culture Phenotyping:
Using PCR which is faster and less expensive to detect Enterococci and vancomycin resistance genes, traditional culture phenotyping could provide more precise information on the cultured species and antibiotic resistance. The authors should consider using both methods in future studies to confirm the results.
3. Data Analysis:
The study used descriptive statistics and chi-square tests to analyze the data. However, logistic regression analysis could provide more robust statistical analysis of the associations between variables.
4. Generalizability:
The study was conducted in Central Italy, and the results may not be generalizable to other regions or populations. It is recommended to consider samples from different regions and populations (from Italy) to confirm the findings.
Round 2
Reviewer 2 Report
The manuscript analyzes the presence of vancomycin-sensitive and vancomycin-resistant Enterococcus in a large number of samples. The subject is interesting but has important methodological limitations that have not been solved by the authors:
1.- The selection of patients is methodologically deficient.
2.- The detection of the microbiome present in the samples would give a truer idea of the situation of this microorganism in this anatomical site.
3.- The search for vancomycin-resistant strains should have been complemented with a selective plate culture, which would have made it pssible to obtain the strain and better study its genome.
Diccionario
